# Arterial Circle of the Brain of the Red-Necked Wallaby (*Notamacropus rufogriseus*)

**DOI:** 10.3390/ani12202796

**Published:** 2022-10-17

**Authors:** Maciej Zdun, Jakub J. Ruszkowski, Maciej Gogulski, Agata Józefiak, Mateusz Hetman

**Affiliations:** 1Department of Animal Anatomy, Poznan University of Life Sciences, Wojska Polskiego 71C, 60-625 Poznan, Poland; 2Department of Basic and Preclinical Sciences, Nicolaus Copernicus University in Torun, Lwowska 1, 87-100 Torun, Poland; 3Department of Preclinical Sciences and Infectious Diseases, Poznan University of Life Sciences, Wołynska 35, 60-637 Poznan, Poland; 4University Centre for Veterinary Medicine, Szydłowska 43, 60-656 Poznan, Poland; 5Centre of Biosciences, Institute of Animal Physiology, Soltesovej 4-6, 040-01 Kosice, Slovakia

**Keywords:** anatomy, vascularization, angiology, marsupials, circle of Willis

## Abstract

**Simple Summary:**

The red-necked wallaby is a small, herbivorous mammal that is native to Australia. In the study, the anatomy of arterial vessels that transport blood to the brain in this species was described and compared with that of other groups of mammals. The results of this research can contribute to further physiological and pathophysiological studies. This is the first description of this anatomical area that has been carried out in a marsupial species.

**Abstract:**

The red-necked wallaby is a medium-sized marsupial species, which have increasingly been kept as pets around the world. In the study, the arterial blood supply for the brain in this species was described. The study was conducted on 50 specimens with two preparation methods. The main artery supplying the brain was the internal carotid artery. The arterial circle of the brain was closed from the caudal side. The anatomy of the arteries of the described region was compared with other groups of mammals. This is the first description of this anatomical area that has been carried out in a marsupial species. Understanding the anatomy of the circulatory system in the wallaby can be valuable for further physiological and pathophysiological studies.

## 1. Introduction

Marsupials are a large and diversified group of mammals. Members of the Macropodoidea, a big marsupial superfamily that contains kangaroos, wallabies, and rat kangaroos, live naturally only in Papua New Guinea and Australia, where they are the main large terrestrial herbivores. The evolution of bipedal locomotion that is typical for macropodoids has affected various morphological and functional adaptations of their cardiovascular, respiratory, and musculoskeletal systems [1,2,3,4]. Wallabies, including the red-necked wallaby or Bennet’s wallaby (*Notamacropus rufogriseus*), are medium-sized hopping, herbivorous marsupials that are native to eastern Australia. 

In captivity, Bennet’s wallabies are kept in European zoos and wildlife parks. Keeping them in zoos dates back to the 1930s, according to the Zoological Society of London [5]. In some countries, they are also considered an alternative to farm animals as a source of meat [6]. Currently, the anatomy of cerebral brain arteries has not been described in any marsupial species, including the red-necked wallaby.

Wallaby medicine is a developing branch of veterinary medicine worldwide [7,8,9,10,11,12,13,14,15]. Among the diseases described in the literature, parasitic infestations [12], bacterial diseases of the oral cavity [13], and cardiomyopathies can be mentioned [8]. Since it is not easy to restrain these species (most of them are kept in extensive enclosures, or are free-roaming), most of the veterinary activities (clinical examination, routine treatment) are carried out with the use of chemical restraint. Furthermore, macropods are exceedingly sensitive to capture myopathy [5]. The increasing number of wallabies kept at zoos, as farm animals, or as pets worldwide, makes it necessary to create diagnostic and treatment protocols for veterinarians. It is important to know the detailed anatomy of every species, in order to acknowledge the pathophysiology of various, often newly discovered diseases. Understanding the anatomy of the brain’s vascular system creates new opportunities for further studies on cardiovascular diseases and their complications. Such diseases have also been described in kangaroos, including red-necked wallabies [14]. Hypertrophic cardiomyopathy may lead to thromboembolic events that affect cerebral vasculature in other mammals [15].

In this study we described, for the first time, the arterial blood supply to the brain of a red-necked wallaby.

## 2. Materials and Methods

### 2.1. Animals

The study was conducted on 50 adult (21 male and 29 female) red-necked wallabies (*Notamacropus rufogriseus*). The analyzed specimens were delivered as post-mortem material from zoological gardens and private breeders in Poland, where animals are maintained under constant veterinary control and anti-parasitic prophylaxis or vaccinations against bacterial diseases, for example, tetanus. All animals that were included in this study had been euthanized [with xylazine 10 mg/kg (intramuscular; i.m.), ketamine 90 mg/kg (i.m.), and pentobarbital 100 mg/kg (intravenous; i.v.)] for medical reasons other than for neurological or cardiovascular disease. The animals had not been euthanized for an experimental purpose. In the study, only animals lacking head and neck trauma of any source were included. All of the animals were frozen prior to the study.

### 2.2. Methods

The methods used in the study were based on traditional anatomical techniques used in angiology research; however, new, high-quality advanced imaging techniques were also used.

The specimens were prepared using three methods, with each specimen being randomly assigned to one of the methods. In twenty-eight specimens, injecting both common carotid arteries with a tinged solution of the chemo-setting acrylic material Duracryl^®^ Plus (SpofaDental, Jičín, Czech Republic) was used. When the material hardened, the specimens were subjected to an enzymatic maceration process (using Persil washing powder). The temperature used for the maceration process was 38 °C, and it lasted 40 days. The final effects of the process were arterial casting of the head and cranial cavity on a skeletal scaffold. The second method, used for 20 specimens, consisted of introducing liquid LBS 3060 latex into the bilateral common carotid arteries. After curing in 5% formalin solution, the specimens were rinsed in water for two days to flush out excess formalin solution. Additional protection was provided with a ventilation system in a room that was used for preparation. The system was set for 15 air changes per hour. The skull bone was cut, carefully, using an oscillating saw, and the skull was opened to enable soft tissue preparation. Then, the blood vessels were prepared manually using surgical instruments. Preparation began with removing skin from the entire neck and head. After this, muscle tissue was gently prepared to avoid damage to the surrounding veins and arteries. Then, prepared arteries were cleaned from excess connective tissue. In this way, blood vessels on the animal’s soft tissues were obtained.

Two cadavers of both genders were used for angio-CT examination. Before the scans, the bilateral common carotid arteries were injected with barium sulphate (barium sulfuricum 1.0 g/mL, Medana, Sieradz, Poland). The heads were fixed motionless on the tomograph’s table. Cone-bean computed tomography (Fidex Animage, Pleasanton, CA, USA) was performed at the University Centre for Veterinary Medicine in Poznan, Poland, with scanning parameters of 110 kVp, 0.08 mAs pet shot, 20.48 mAs (Total mAs), and a reconstructed slice thickness of 0.29 mm. After the examination scan was reconstructed, the 3D model was created using public available software 3D Slicer (version 5.0.3). All of the excess tissue, except for bone tissue and arteries, was cut out, digitally. The bone tissue and arteries were digitally colored—yellow for bone and red for arteries.

The names of the anatomical structures were standardized according to Nomina Anatomica Veterinaria [16].

For the next process, all of the preparations were photographed and described in detail. The described material was then compared to those of other groups of mammals. A digital camera (Nikon D3200) was used to obtain the photographs. The photographs were saved in JPG format. GIMP v2.10.18, digital image editing software, was used to process the photographs.

## 3. Results

Blood enters the cranial cavity via the internal carotid artery (*arteria carotis interna*), which is a branch of the common carotid artery (*arteria carotis communis*) (Figure 1).

There was no carotid sinus (*sinus caroticus*) found in the initial course of the internal carotid artery. The course of the internal carotid artery was straight, and it did not turn sideways. On its extracranial part (from branching off of the common carotid artery to enter the cranial cavity), it was not providing any branches, and was not anastomosing with other arterial vessels. This artery penetrated the cranial cavity and laid on the base of the cerebrum (Figure 2), moving towards the optic canal.

After passing through it, the internal ophthalmic artery (*arteria ophthalmica interna*), was observed to enter the orbit. The internal carotid artery was a strong vessel over its entire length. It divided into the internal ophthalmic artery and the rostral cerebral artery (*arteria cerebri rostralis*) (Figure 1 and Figure 2). At that point, the diameters of these vessels became smaller. The rostral choroidal artery (*arteria choroidea rostralis*), which runs dorsally between the temporal lobe of the brain and the cerebral crus, branched off from the beginning segment of the rostral cerebral artery. The rostral cerebral artery gave way to the middle cerebral artery (*arteria cerebri media*), a vessel with a large diameter (Figure 2). It ran dorsally, and surrounded the piriform lobe from its rostral side, and divided into numerous branches. The middle cerebral artery branched off into four main branches (Figure 3).

From the beginning section of the middle cerebral artery branched off two vessels: the frontal and the orbital branches. Altogether, those vessels were observed to branch off together as a common trunk. The frontal branch divided further into two smaller vessels once again: the ventral frontal branch and the dorsal frontal branch. The caudal olfactory artery branched off near the point where the above-mentioned common trunk branched off. Next, in the caudal direction, branched off temporal branches (divided secondarily into the middle and ventral temporal branches). Finally, the middle cerebral artery divided into the parietal branch (divided secondarily into the rostral and caudal parietal branches) and the dorsal temporal branch. The dorsal temporal branch was the vessel observed to have the largest diameter. Once the middle cerebral artery branched off, the rostral cerebral artery’s lumen reduced significantly. In 39 specimens, the rostral cerebral artery branched off the rostral communicating artery (*arteria communicans rostralis*). This vessel branched off at a short distance from the middle cerebral artery. The presence of the rostral communicating artery provided evidence that the arterial circle of the brain is closed on the rostral side. The rostral cerebral artery was observed to head rostrally, and then change direction, laying in the longitudinal fissure of the brain. Between the olfactory bulbs, the internal ethmoidal artery (*arteria ethmoidalis interna*) branched off, leading to the cribriform plate of the ethmoid bone.

The caudal communicating arteries (*arteriae communicans caudales*), with the basilar artery (*arteria basilaris*) together creates the caudal part of the arterial circle of the brain (Figure 4).

In the vicinity of the origin of the caudal communicating arteries, the internal carotid artery gives way to the caudal cerebral artery (*arteria cerebri caudalis*) (Figure 5).

In 20 specimens, these vessels were found to branch off with a common trunk. The caudal cerebral artery is relatively weak, and supplies a small, caudally located area of the brain. From the caudal communicating arteries branched off the rostral cerebellar artery (*arteria cerebelli rostralis*) (Figure 4), supplying the anterolateral part of the cerebellum. Near the middle of the length of the basilar artery, the caudal cerebellar artery (*arteria cerebelli caudalis*) branched off. This vessel supplies the caudal part of the cerebellum. In addition, small arteries leading to the medulla oblongata were observed to branch off from the basilar artery. The basilar artery is an odd vessel with the same diameter over its entire length (Figure 6), and caudally joins the vertebral arteries (*arteriae vertebrales*).

## 4. Discussion

The samples collected from fifty red-necked wallabies were investigated for the arterial blood supply to the brain. In the analyzed species, there is a strong internal carotid artery, the main blood source for the brain. Similar observations have been made in the representatives of the Carnivora: domestic dog [17,18], dingo, arctic fox, red fox, pale fox [19], and silver fox [20,21]; *Perissodactyla*: horse [17,22], Przewalski’s horse, Grévy’s zebra, Grant’s zebra, Mongolian khulan, donkey, and South American tapir [22]; lagomorphs: European rabbit [23] and European brown hare [24]; as well as Old and New World camels: Bactrian camel, guanaco, lama [25], and dromedary camel [25,26,27]. Among rodents, only some of the described species were found to have an internal carotid artery: agouti [28], Mongolian gerbil [29], European beaver [30], Egyptian spiny mouse [31], and rat [32]. In Habermehl’s [18] description of this vessel, there is information about the presence of a carotid sinus in the dog. This is a thickening of 3–4 mm in size, at the beginning of this vessel. The presence of this structure has also been described in the cat [33], despite the absence of an internal carotid artery in adults of this species. In other animals with a preserved internal carotid artery, there is no information on the occurrence of the carotid sinus. Similarly, in the analyzed red-necked wallaby, the occurrence of this structure was not observed. The straight course of the internal carotid artery, as observed in the red-necked wallaby, has been previously described in a horse [34]. In camelids and canines, this vessel’s course is sinuous. In the dog, it creates a pronounced vascular loop at the height of the rostral foramen of the carotid artery canal, changing its direction 180 degrees [17,25]. In some domestic cats’ fetuses and juvenile domestic cats, the connection between the internal carotid artery and the pharyngeal artery has been described. This anastomosis has been observed to be very close to the base of the skull. In adult domestic cats, the internal carotid artery obliterates, and after this, the connection of this vessel to the arteries at the base of the brain no longer exists [33]. The results of the final division of the internal carotid artery are two branches: the rostral cerebral artery, and the internal ophthalmic artery. In some rodent species, the internal ophthalmic artery has been found to have a large diameter [35,36].

The vessels branching off from the arterial circle of the brain show some characteristics of the analyzed species. In the red-necked wallaby, it was confirmed that four main branches branched off from the middle cerebral artery. Skoczylas et al. [37] found three main branches in the polar fox, which he named the rostral middle cerebral artery, dorsal middle cerebral artery, and the caudal middle cerebral artery. The domestic pig and wild boar have shown significant variations in the branching off the vessels from the middle cerebral artery. In some individuals, the division of the middle cerebral artery occurred in the dorsally located point, while in others there was found a double middle cerebral artery [38,39]. The dorsal temporal branch was the strongest in the sheep, polar fox, and silver fox [21,37,40].

In most of the analyzed specimens, the rostral communicating artery was present. In a study that was conducted by König [41] on domestic cattle, the arterial circle of the brain was open on the rostral side in the majority of the cases. An open arterial circle of the brain was also found in representatives of rodents, such as in the guinea pig [36,42], nutria [43], the ground squirrel [44], and the chinchilla [45].

In the described species, a strongly developed internal ophthalmic artery was observed. This vessel was an elongation of the internal carotid artery. Usually, the internal ophthalmic artery is a tiny vessel [17]. The well-developed internal ophthalmic artery has been previously observed in the guinea pig [36]. In this species, the connection of the previously mentioned vessel to the arterial circle of the brain was similar to the pattern found in the red-necked wallaby; however, the vessel originated from the caudal communicating artery near the middle cerebral artery because of the absence of the internal carotid artery. Among Carnivora, the internal ophthalmic artery is also well-developed in the American mink [19].

The caudal cerebral artery in the carnivores, domestic dog [17,18], dingo, arctic fox, red fox, pale fox [19], and silver fox [20,21]; in *Perissodactyla*, horse [17,18], Przewalski’s horse, Grévy’s zebra, Grant’s zebra, Mongolian khulan, donkey, and South American tapir [22]; in lagomorphs, European rabbit [23] and European brown hare [24]; and even-toed ungulates, Bactrian camel, guanaco, lama [25], dromedary camel [25,26,27], domestic cattle, banteng, yak, American bison, and European bison [46], branched off from the caudal communicating arteries. In the red-necked wallaby, in most cases, this vessel branched off from the internal carotid artery. Similar observations have been made in some rodents with the internal carotid artery. This mode of caudal cerebral artery departure has been described in the Egyptian spiny mouse [31], Mongolian gerbil [29], and only parts of rats [47]. In some rodents with the internal carotid artery, such as the European beaver or agouti, the caudal cerebral artery branched off from the terminal branches of the basilar artery [28,30].

Although the red-necked wallaby’s arterial circle of the brain is closed on the caudal side, unlike the Mongolian gerbil, areas that are supplied by the basilar artery and internal carotid artery can be seen [29]. In the Mongolian gerbil, the caudal communicating arteries are not present. This makes it clear that the basilar artery supplies the midbrain and hindbrain, and the internal carotid artery supplies the forebrain and interbrain. In the red-necked wallaby, admittedly, there are the caudal communicating arteries, but their diameters are similar to the basilar artery. These vessels are definitely smaller in diameter than that of the internal carotid artery. This leads us to believe that blood flowing to the midbrain and hindbrain comes from both the internal carotid artery and the basilar artery, and that the basilar artery is not involved in supplying blood to the forebrain and interbrain. The basilar artery is the only source of blood to the cerebellum in some rodents, such as ground squirrels [44]. In the chinchilla, nutria, and the porcupine, the basilar artery creates an arterial circle of the brain alone [35,43,45,48]. In those species, the caudal communicating arteries are strong vessels, and the internal carotid artery does not exist, or is highly reduced, hence not supplying blood to arteries at the base of the brain. Baldwin and Bell [49,50,51] pointed out the effect of reduced vessel diameter on blood delivery to a given vascular area. Their series of studies on ruminants proved that reducing the lumen of the basilar artery in the caudal segment results in blood flowing from the circle to the basilar artery, and not vice versa. In this group of animals, despite the basilar artery not transporting blood to the brain, the extracranial segment of the internal carotid artery obliterates [46]. Therefore, blood is not transported to the brain from vertebral arteries or the common carotid artery, which is exactly the opposite to the red-necked wallaby. In ruminants, the brain is supplied with blood by arteries that are derivatives of the maxillary artery—caudal branch to the rostral epidural rete mirabile and rostral branches to the rostral epidural rete mirabile [46]. This vascular pattern creates a necessity for the rostral epidural rete mirabile to be present in this group of mammals.

This study complements the anatomical research on brain vascularization among various species with the red-necked wallaby. Differences in the arterial blood flow in the described area may be related to the phylogenetic distance between marsupials and the other above-mentioned mammal species. Additionally, it may be important for anatomists, physiologists, and veterinarians to know how to describe in detail the vascular pattern of the arteries that supply blood to the brain, which differs significantly between various groups of mammals.

## 5. Conclusions

This study provided detailed information about the brain vascularization of the red-necked wallaby using three methods of specimen preparation, including advanced imaging with cone-bean computed tomography. In the described species, the main blood vessel supporting the brain is an internal carotid artery. Moreover, the arterial circle of the brain is closed on the caudal side. In some of the described preparations, independent branching of the separate vessels from the internal carotid artery was seen: the caudal cerebral artery and the caudal communicating artery. The internal ophthalmic artery is the well-developed elongation of the internal carotid artery. Obtained findings were compared with other mammal species’ brain blood flow, described in scientific literature. The results could be useful for further research about wallaby physiology and medicine, due to its increasing popularity among breeders and private holders. Since various cardiovascular diseases have been described before in this species, a detailed description of angio-anatomical structures may contribute to establishing diagnostic protocols, and help in pathophysiological studies.

## Figures and Tables

**Figure 1 animals-12-02796-f001:**
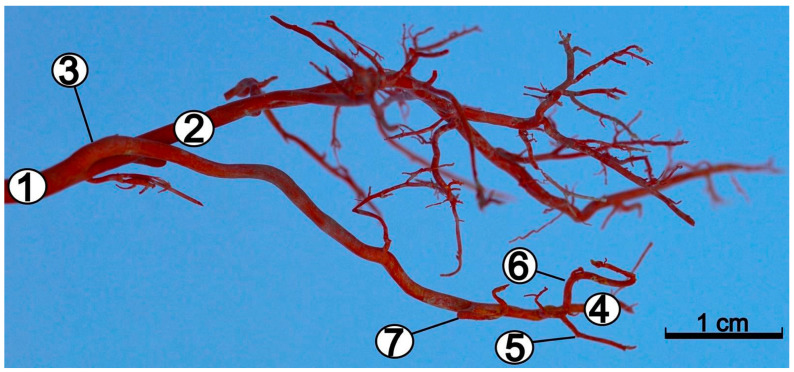
Arteries of the left side of the head of the red-necked wallaby. Corrosion cast. 1—common carotid artery; 2—external carotid artery; 3—internal carotid artery; 4—internal ophthalmic artery; 5—rostral cerebral artery; 6—middle cerebral artery; 7—caudal communicating artery.

**Figure 2 animals-12-02796-f002:**
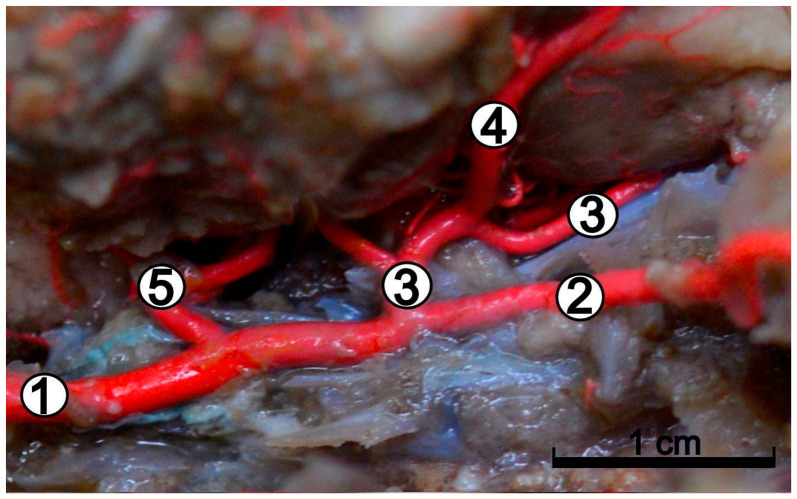
Branches of the internal carotid artery in the cranial cavity between the brain and the base of the skull. Latex preparation. 1—internal carotid artery; 2—internal ophthalmic artery; 3—rostral cerebral artery; 4—middle cerebral artery; 5—caudal communicating artery.

**Figure 3 animals-12-02796-f003:**
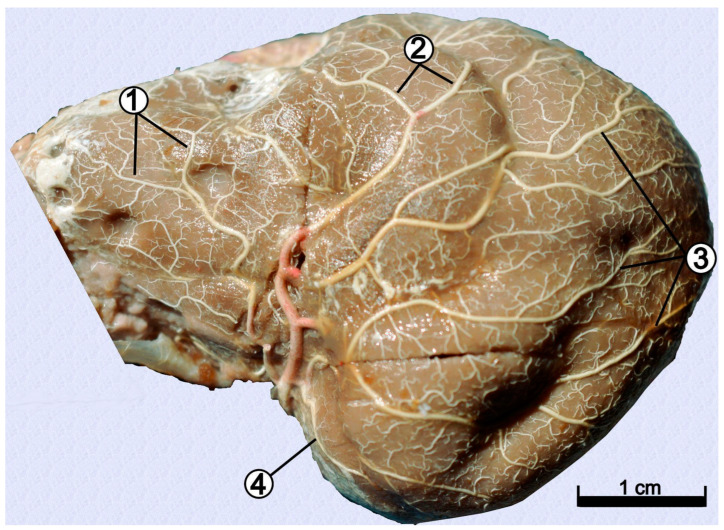
Lateral view of the left cerebral branches of a middle cerebral artery. Latex preparation of red-necked wallaby brain. 1—frontal branches; 2—parietal branches; 3—temporal branches; 4—caudal olfactory artery.

**Figure 4 animals-12-02796-f004:**
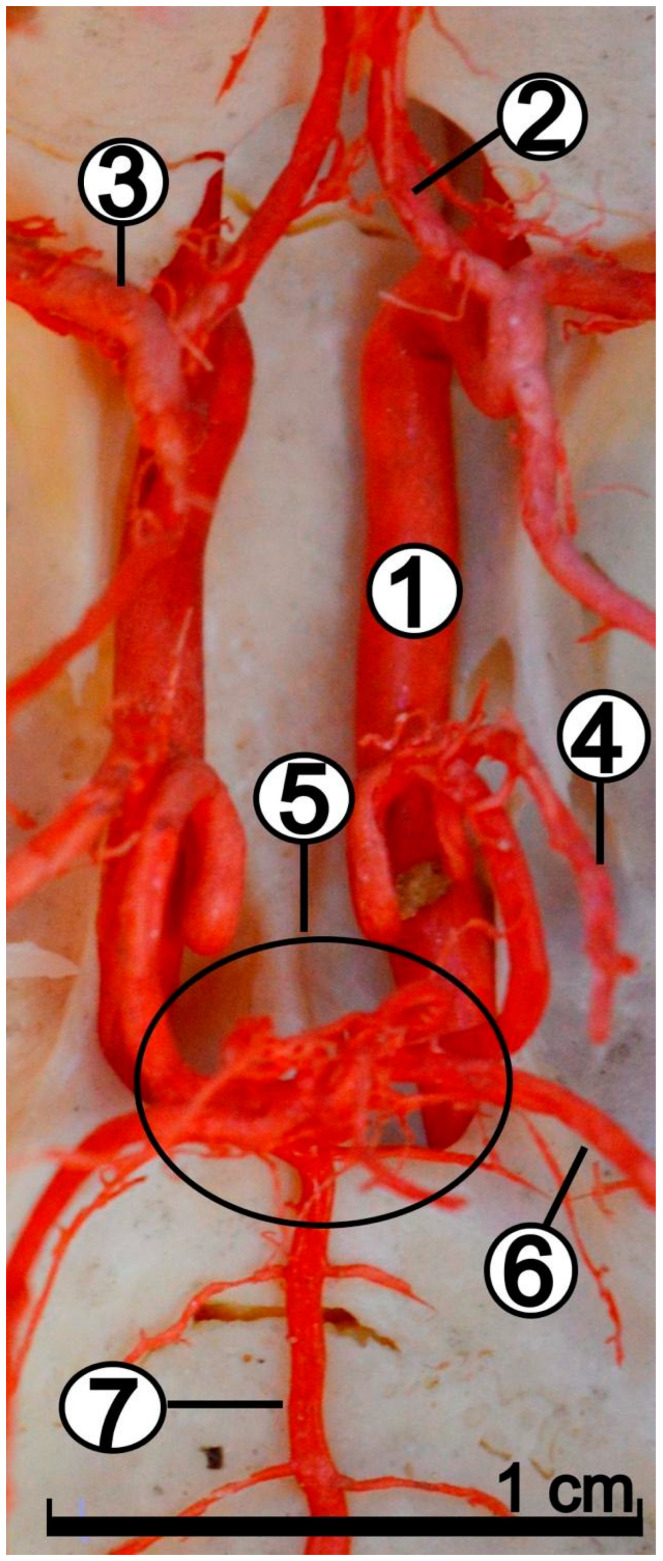
Dorsal view of the arterial circle of the brain. Vessels in the cranial cavity on the skeletal scaffold. Corrosion cast. 1—internal carotid artery; 2—rostral cerebral artery; 3—middle cerebral artery; 4—caudal cerebral artery; 5—caudal communicating artery; 6—rostral cerebellar artery; 7—basilar artery.

**Figure 5 animals-12-02796-f005:**
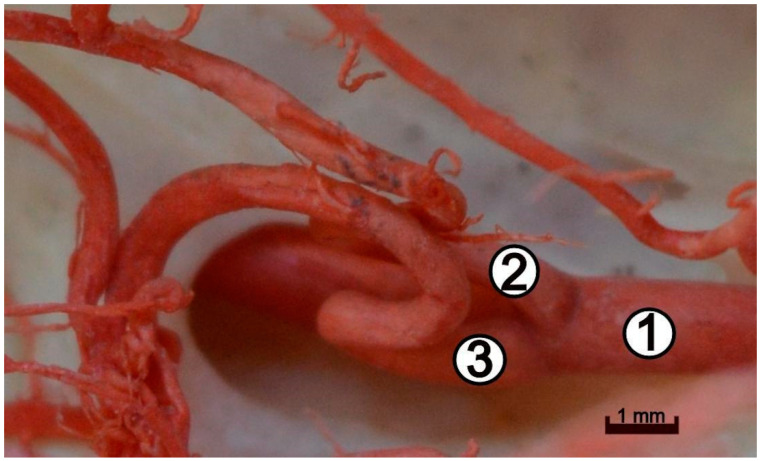
The left side of the caudal part of the arterial circle of the brain. Branches of the internal carotid artery, close up. Independent branching of the caudal cerebral artery and caudal communicating artery from the internal carotid artery. Corrosion cast. 1—internal carotid artery; 2—caudal cerebral artery; 3—caudal communicating artery.

**Figure 6 animals-12-02796-f006:**
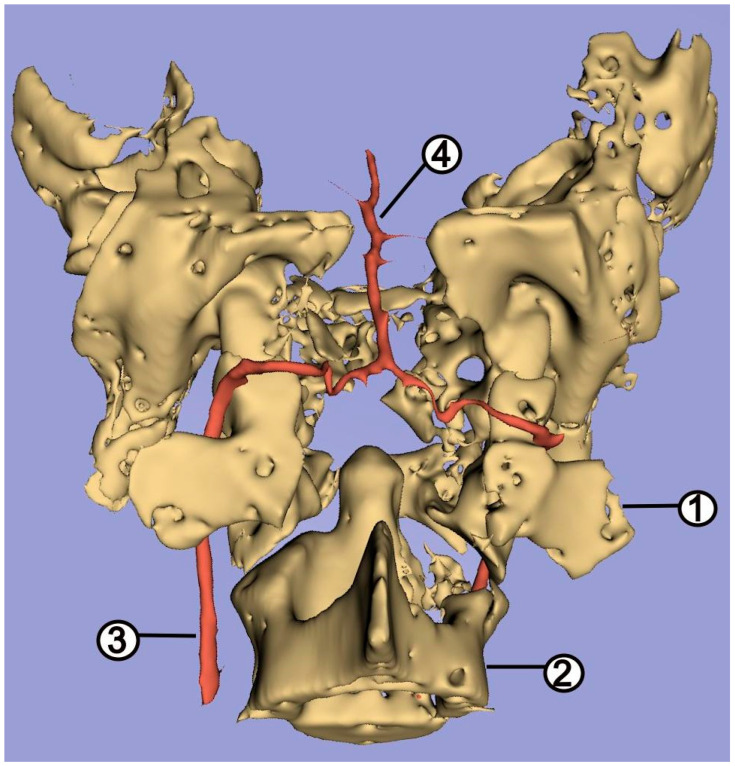
3D reconstruction of the basal artery on bone scaffolding. 1—first cervical vertebra; 2—second cervical vertebra; 3—vertebral artery; 4—basilar artery.

## Data Availability

The data presented in this study are available on request from the corresponding author.

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
