# Peer review of "Arterial Circle of the Brain of the Red-Necked Wallaby (Notamacropus rufogriseus)"

_animals, 2022, doi:10.3390/ani12202796_

Round 1
Reviewer 1 Report
Authors describe the basilar artery with the same diameter over its entire length similar to diameter of the caudal communicating arteries. Supporting this observation, it would be a good idea to show an appropriate vascular specimen. Figure number 4 shows that this blood vessel looks as reducing its diameter from the circle to the caudal direction. It would suggest the caudal direction of blood flow as suggested by Baldwin and Bell (1963). Analyzing the role of the basilar artery (discussion), I propose to include information about the sole participation of the basilar artery in the formation of the arterial circle of the brain, e.g., nutria or chinchilla.
Author Response
The previous figure had a methodological error we missed – the artery did not fill entirely, which looked like it was reducing its diameter in a caudal direction. The figure was corrected. Additionally, we added 3D reconstruction based on a cone-bean computed tomography scan for another proof of the stable diameter of the artery.
The information on the sole participation of the basilar artery in the formation of the arterial circle of the brain in nutria and chinchilla was added.
Reviewer 2 Report
The present paper is a very good description of the anatomy of brain the arterial vessels. The study was conducted in 48 adults red-necked wallaby, which are enough for its purposes. It is well written and easy to follow. The results show very good quality images, and the discussion is also interesting as authors make a good comparison between different mammals species, including rodents. Surely, what would improve the results is to provide an imaging study of the same arterial blood vessels, which could have been done before euthanizing some of the animals. For example, radiographies and angiographies would be a good complement to this study.
Author Response
Figure 6 was added (3D reconstruction based on angiograph made with a cone-bean computed tomography.
Reviewer 3 Report
Authors present the basal artery as a vessel with the same diameter over its entire length. Nevertheless, the basal artery in Figure 4 clearly narrows in the caudal direction.
Author Response
The previous figure had a methodological error we missed – the artery did not fill entirely, which looked like it was reducing its diameter in a caudal direction. The figure was corrected. Additionally, we added 3D reconstruction based on a cone-bean computed tomography scan for another proof of the stable diameter of the artery.